# Involvement of Nrf2 Activation and NF-kB Pathway Inhibition in the Antioxidant and Anti-Inflammatory Effects of Hesperetin in Activated BV-2 Microglial Cells

**DOI:** 10.3390/brainsci13081144

**Published:** 2023-07-29

**Authors:** Jasmine A. Evans, Patricia Mendonca, Karam F. A. Soliman

**Affiliations:** 1Division of Pharmaceutical Sciences, College of Pharmacy and Pharmaceutical Sciences, Institute of Public Health, Florida A&M University, Tallahassee, FL 32307, USA; jasmine2.evans@famu.edu; 2Department of Biology, College of Science and Technology, Florida A&M University, Tallahassee, FL 32307, USA

**Keywords:** hesperetin, Nrf2, Keap1, oxidative stress, NF-kB signaling, PD-L1, Alzheimer’s disease

## Abstract

Alzheimer’s disease is a progressive neurodegenerative disorder leading to cognitive decline and memory loss. The incidence of this disease continues to increase due to the limited number of novel therapeutics that prevent or slow down its progression. Flavonoids have been investigated for their potential effects on cellular damage triggered by excessive reactive oxygen species (ROS) and neuroinflammatory conditions. This study investigated the effect of the flavonoid hesperetin on LPS-activated murine BV-2 microglial cells. Results show that hesperetin reduced nitric oxide levels and increased catalase, glutathione, and superoxide dismutase levels, suggesting its potential to reduce neuroinflammation and oxidative stress. Moreover, RT-PCR arrays showed that hesperetin modulated multiple genes that regulate oxidative stress. Hesperetin downregulated the mRNA expression of *ERCC6*, *NOS2*, and *NCF1* and upregulated *HMOX1* and *GCLC*. RT-PCR results showed that hesperetin-induced Nrf2 mRNA and protein expression in LPS-activated BV-2 microglial cells is involved in the transcription of several antioxidant genes, suggesting that hesperetin’s antioxidant effects may be exerted via the Keap1/Nrf2 signaling pathway. Furthermore, the data demonstrated that hesperetin reduced the gene expression of PD-L1, which is upregulated as an individual ages and during chronic inflammatory processes, and inhibited the expression of genes associated with NF-kB signaling activation, which is overactivated during chronic inflammation. It was concluded from this investigation that hesperetin may have therapeutic potential to prevent or slow down the progression of neurodegenerative diseases, such as Alzheimer’s disease, by reducing chronic oxidative stress and modulating neuroinflammation.

## 1. Introduction

The onset of neurodegenerative diseases has continuously increased due to increased life expectancy. In the U.S., the number of individuals aged sixty-five and older with neurodegenerative diseases is predicted to reach over thirteen million by the end of 2023 because of the limited number of medications to attenuate or prevent neurodegenerative conditions such as Alzheimer’s disease (A.D.) [1]. Slowing the progressive loss of neurons could counteract the serious health threat neurodegenerative diseases have presented in the geriatric population [2,3,4,5]. A.D. causes a decline in memory and learning and accounts for 80% of dementia cases diagnosed [6,7,8]. The pathological characteristics of A.D. include the intracellular destabilization of tau protein leading to neurofibrillary tangles and the accumulation of amyloid-beta (Aβ) extracellularly causing senile plaques, which are partly thought to be responsible for the onset of A.D. [9,10]. Although the etiology of A.D. has not been elucidated, aging, chronic oxidative stress, and inflammation have been indicated as common causes. Aging is an irreversible process that makes the brain prone to impaired self-repair abilities and leads to oxidative stress because of an imbalance in the production of reactive oxygen species (ROS), free radicals, and antioxidants [4,5]. Understanding the mechanisms linked to chronic oxidative stress, aging, neuroinflammation, and the central nervous system (CNS) may be critical in preventing and/or slowing the progression of A.D. and other neurodegenerative conditions.

Currently, no cure or therapy is targeted at both hallmarks to slow A.D. progression. However, it is known that increased oxidative stress causes changes in antioxidant levels in tissues, enhances mitochondrial dysfunction, and may change metal homeostasis, leading to additional production of ROS [7]. Oxidative stress is a critical factor contributing to the chronic activation of microglial cells in the brain [11]. Normally, microglial cells protect the nervous system by destroying pathogens, eliminating debris, and enhancing the immune response [12]. Microglial cells are usually at rest in physiological conditions and serve as host defenses, performing immune surveillance [13]. Once activated, the microglia undergo morphological changes that induce an array of surface receptors and release multiple soluble factors [14]. The excessive accumulation and production of Aβ protein is another factor that leads to the chronic activation of microglial cells, leading to the overproduction of inflammatory mediators, which could induce neuronal cell damage [2,3]. Studies suggest that the continued activation of microglial cells via oxidative stress and neuroinflammation contributes to the onset of neurodegenerative diseases, specifically A.D. [2].

Typically, the mechanisms that protect the cells against oxidative stress include the endogenous antioxidant pathways that boost cytoprotective enzyme expression, which can scavenge free radicals and decrease cellular injury caused by excessive ROS [8]. A major pathway that maintains redox homeostasis in the brain is the transcription factor nuclear erythroid 2-related factor 2 (Nrf2), which binds to the antioxidant response element (ARE) located at the promoter region of antioxidant genes [8,9]. Nrf2 is ubiquitously expressed in the brain and is a crucial defense mechanism against oxidative stress toxicity [15]. This transcription factor upregulates several antioxidant enzymes, increasing the expression of anti-inflammatory mediators and initiating mitochondrial signaling pathways [16]. It is critically important to have crosstalk between antioxidant and anti-inflammatory pathways, which is considered a secondary effect of Nrf2 antioxidant activity [17]. Nrf2 is normally bound to Kelch-like E.C.H.-Associated Protein 1 (Keap1), which targets it and causes it to degrade via the proteasomes [8,18,19,20,21,22]. In oxidative stress, Keap1 functional groups are modified, resulting in conformational changes that reduce Keap1 and degrade Nrf2, which allows it to accumulate in the cell’s nucleus [15]. Nrf2, a master regulator of cellular redox homeostasis, decreases as a person ages [23]. This reduction has been linked to increased expression of negative regulators, leading to reduced protein expression and decreased responsiveness of the Nrf2 signaling pathway [22].

Moreover, brain inflammation is an important defense mechanism against infection, injury, and toxins in normal physiological conditions and in acute situations. However, disrupting the homeostasis of inflammatory and anti-inflammatory factors results in chronic inflammation, as seen in A.D. [24,25]. Microglia are activated in response to chronic neuroinflammation, which initiates the release of multiple mediator factors. A critical immune checkpoint upregulated in inflammation is programmed cell death 1(PD-1), whose expression has been reported on microglial cells [26,27]. This transmembrane protein is engaged by programmed death ligand 1 (PD-L1), resulting in the dephosphorylation and deactivation of effector proteins [28]. This pathway has been extensively investigated in several types of cancer. Still, few studies on its involvement in A.D. are available. Kummer et al. reported that the modulation of the PD-1/PD-L1 pathway is significant in the phagocytosis of Aβ, which is one of the hallmarks of A.D. [29]. Additionally, the activation of nuclear factor-kB (NF-kB), which is a family of transcription factors, is suggested to be the primary regulator for inflammatory responses [30]. This family comprises NF-kB1, NF-kB2, RelA, and other proteins mediating the transcription of pro-inflammatory genes [31]. NF-kB activation involves canonical and non-canonical pathways, both critical in regulating inflammatory responses through different mechanisms [32]. These pathways have been widely studied because they are crucial in the inflammatory response and have been proven to be over-activated in neurodegenerative diseases linked to chronic inflammation. Therefore, there is a need to identify a natural compound that can decrease inflammatory gene expression, such as those associated with NF-kB signaling, and increase the expression of Nrf2 as a potential target for slowing the onset and progression of neurodegenerative diseases, such as A.D.

Previous studies have suggested natural compounds like flavonoids may prevent cellular injury and slow cellular oxidative stress and inflammation [33]. Flavonoids, polyphenolic metabolites found in fruits, herbs, and vegetables, have a variety of medicinal applications, including activating antioxidant enzymes [34]. Flavonoids have been classified based on their scavenging activity against free radicals and antioxidant activity. Moreover, it has been suggested that numerous flavonoids impede neurotoxic pathways associated with neurodegeneration [14,15]. Hesperetin (HPT) (Figure 1) is a derivative of the naturally occurring flavonoid hesperidin from the *Citrus* L. plant, found in citrus fruits such as oranges, lemons, and mandarins. HPT is an aglycone glycoside of hesperidin, which widely exists in Chinese medicinal herbs [17] and has various pharmacological activities, including neuroprotective, anti-inflammatory, and antioxidative properties [35,36,37]. HPT’s Other biological properties include its effects on cancer, diabetes, and cardiovascular disease. HPT may be ideal for neurodegenerative treatment because its small molecular weight suggests it crosses the blood-brain barrier [2,17]. It is also critical to mention that HPT is more bioactive than hesperidin, which makes its absorption more effective in the gastrointestinal tract. However, HPT’s antioxidant properties and cellular mechanisms have not been investigated in models associated with neurodegeneration.

Since oxidative damage and chronic inflammation are critical to the progression of dementia and neuronal loss, in the present study we investigated the molecular mechanisms of HPT’s antioxidant and anti-inflammatory effects on LPS-activated murine BV-2 microglial cells, a well-established model to investigate neurodegeneration. The goal was to explore HPT molecular mechanisms in the modulation of oxidative stress and the activation of the Nrf2/Keap1 signaling pathway, as well as the HPT modulatory effect on the inflammatory process through the regulation of NF-kB and PD-1/PD-L1 signaling pathways.

## 2. Materials and Methods

### 2.1. Cell Lines, Chemicals, and Reagents

Murine BV-2 microglial cells were provided by Dr. Elizabetta Blasi [38]. The Dulbecco’s Modified Eagle’s Medium (DMEM) high glucose, fetal bovine serum heat inactivated (FBS-HI), and penicillin/streptomycin were obtained from Genesee Scientific (San Diego, CA, USA). Dimethyl sulfoxide (DMSO) and resazurin were bought from Sigma-Aldrich Co. (St. Louis, MO, USA). Catalase (Item # 707002), glutathione (Item # 703002), and superoxide dismutase (Item # 706002) assay kits and HPT (item # 1006084) were obtained from Cayman Chemical (Ann Arbor, MI, USA). PCR arrays (item #10034391), iScript advanced reverse transcriptase kit (item # 1725038), Universal SYBR green supermix (item # 1725261), mouse primer PCR pathway oxidative stress and antioxidative defense arrays (CAT # 10034391), and PCR primers were bought from Bio-Rad (Hercules, CA, USA). Each reagent and plate for western analysis were purchased from ProteinSimple (San Jose, CA, USA). Primary antibodies were obtained from Cell Signaling (Danvers, MA, USA).

### 2.2. Cell Culture

Murine BV-2 microglial cells were grown in DMEM with 10% FBS and 1% penicillin (100 U/mL)/streptomycin (0.1 mg/mL) and incubated in 5% CO_2_ and 37 °C. Before each assay, cells were subcultured and grown to 90% confluency in flasks (T-75). Media used for plating each experiment consisted of DMEM with 2.5% FBS and without penicillin/streptomycin.

### 2.3. Cell Viability

Resazurin assay was utilized for BV-2 microglial cell viability assessment. Then, 96-well plates were prepared with a density of 3 × 10^4^ cells (100 μL/well) and incubated nightly in media with DMEM, 2.5% FBS, and without penicillin/streptomycin. The following day, the cells were treated with control (media only), control (cells + DMSO), or distinct concentrations of HPT (0.78–200 μM). HPT was dissolved in DMSO before adulteration in the media, and the final DMSO concentration did not exceed 0.1% [30]. HPT effects were measured after a 24 and 48 h incubation period for cell viability. The resazurin solution (0.5 mg/mL) in a volume of 20 μL was added to the plate and incubated again for 4 h. Quantitative analysis of resazurin conversion was measured at an excitation/emission of 550/580 nm wavelengths using a microplate reader, the Infinite M200 (Tecan Trading AG, Männedorf, Switzerland). Alive cells reduced resazurin to resorufin, resulting in fluorescence modifications.

### 2.4. Nitric Oxide (Griess) Assay

Nitric oxide (NO) production of the BV-2 microglial cells was determined by measuring the amount of nitrate (NO_2_^−^), a relatively stable oxidation product, released in the cell culture supernatant. The Griess reagent (1% sulfanilamide and 0.1% N-(1-naphthyl)—ethylenediamine hydrochloride in 5% phosphoric acid (H_3_PO_4_)) which is a colorimetric assay, was used. The BV-2 microglial cells (3 × 10^4^ cells/well in a 96-well plate) were seeded overnight to attach, then treated as described previously for 24 h. An equal volume of the Griess reagent was combined with cell culture supernatant and incubated for 10 min at room temperature, covered from light. Absorbance at 550 nm was measured with a microplate spectrophotometer. The concentration of nitrite in the samples was calculated using the standard curve generated by sodium nitrite (NaNO_2_) prepared freshly in the culture medium, subtracting the background nitrite.

### 2.5. Catalase, Superoxide Dismutase, and Glutathione Assays

*Cell Preparation:* The cells were treated with LPS (1 µg/mL) and HPT (100 µM) alone and then treated with the combination of HPT (100 µM) and LPS (1 µg/mL) for 24 and 48 h. Cells were then harvested using the scrapper, collected, and centrifuged at 1000× *g* for 3 min. The cell pellet was sonicated in 1 mL of cold buffer at a pH of 7.2, containing 1 mM of EDTA and 50 mM of potassium phosphate. Homogenized cells were then centrifuged for 15 min at 10,000× *g*, and the supernatant was removed and stored on ice.

Catalase Assay (CAT): kits were bought from Cayman Chemical (item # 707002). Supplied reagents included catalase assay buffer, catalase sample buffer, catalase formaldehyde standard, catalase control, catalase potassium hydroxide, catalase potassium peroxide, catalase purpald, and catalase potassium periodate. These reagents were prepared according to the provided protocol. Briefly, 100 μL of diluted assay buffer, 30 µL of methanol, and 20 µL of the standard or the sample were added to each designated well. The total volume was 240 µL. The reaction was initiated by adding diluted hydrogen peroxide. The reaction plate was covered and incubated on the shaker for 20 min at room temperature. Reaction termination was achieved by adding 30 µL of potassium hydroxide and 30 µL of catalase purpald to each well. The plate was then covered and incubated at room temperature on the shaker. A 10 µL of catalase potassium periodate was added, and the plate was covered and incubated for five minutes on the shaker at room temperature before measuring each well’s absorbance at 540 nm using the microplate reader.

Superoxide Dismutase Assay (SOD): kits were purchased from Cayman Chemical (item # 706002). Supplied reagents included assay buffer, sample buffer, radical detector, SOD standard, and xanthine oxidase, all prepared according to the provided protocol. A total of 200 µL of the diluted radical detector and 10 µL of the standard or sample were added to designated wells on the plate. The reaction was started by adding 20 µL of diluted xanthine oxidase to all wells. The plate was then carefully tapped for a few seconds to mix the solution, and then covered with the plate cover. The plate was incubated on a shaker for 30 min at room temperature, and the absorbance was read at 450 nm using the microplate reader.

Glutathione Assay (GSH): kits were purchased from Cayman Chemical (item # 703002). The supernatant was removed and stored on ice to conduct deproteination before assaying of sample or standard were added to the plate and covered. Deproteination required the addition of an equal volume of newly prepared metaphosphoric acid (MPA) reagent. The MPA reagent was prepared by dissolving 5 g in 50 mL of HPLC-grade water, then adding the supernatant of the samples and mixing by vortexing. The mixture stood at room temperature for 5 min before centrifugation at 2000× *g* for 2–3 min. The supernatant of each sample was then collected without disturbing the precipitate and transferred to a fresh tube. Triethanolamine (TEAM) reagent (531 μL of triethanolamine with 469 μL of water), stable for 4 h, was added to the samples (50 μL per mL of supernatant) and mixed by vortexing. The samples were then ready for measurement. The standard curve was prepared according to the serial dilution instructions provided by the manufacturer. The assay cocktail was prepared by mixing in a 20 mL tube: 11.25 mL M.E.S. Buffer, 0.45 mL of reconstituted cofactor mixture, 2.1 mL reconstituted enzyme mixture, 2.3 mL water, and 0.45 mL reconstituted 5,5′-dithio-bis-2-nitrobenzoic acid (DTNB). A 150 μL assay cocktail was added immediately to each well for 200 μL per well. The plate cover was placed and incubated in the dark on a shaker for 25 min at room temperature before measuring the absorbance at 410 nm using a microplate reader.

### 2.6. Real-Time Polymerase Chain Reaction (RT-PCR) Arrays

BV-2 cells were treated with HPT (100 µM), LPS (1 µg/mL), or a combination of HPT (100 µM) and LPS (1 µg/mL). After 48 h, cells were harvested using the scraper, media aspirated, and cells rinsed twice with PBS. Then cell pellets were lysed with 1 mL TRIzol reagent. Chloroform (0.2 mL) was added to samples, shaken, and incubated for 2–3 min at 15–30 °C, then centrifuged at 10,000× *g* for 15 min at 2–8 °C. The aqueous phase was transferred and mixed with 0.5 mL isopropyl alcohol for RNA precipitation. After incubation, samples were centrifuged and the supernatant removed, then RNA pellets were washed with 75% ethanol. Samples were vortexed before being centrifuged at 7500× *g* for 5 min at 2–8 °C. The ethanol was removed, and the RNA pellet was dried, reconstituted in RNase-free water, and incubated for 30 min on ice. Nanodrop (Thermo Fischer Scientific, Wilmington, DE, USA) was used to measure RNA concentration and quantity. The cDNA was synthesized from the mRNA using iScript advanced reverse transcriptase kit from Bio-Rad. The iScript advanced reaction mix (containing primers), reverse transcriptase, the sample, control assay template, and water were added to the 0.2 mL tube. The thermal-cycling program included two steps: samples were first incubated, amplified, and performed following the manufacturer’s protocol (Bio-Rad). The sample (200 ng cDNA/reactions), master mix (Universal SYBR Green Supermix (item # 1725261)), primer, and nuclease-free water were mixed into each well. The thermal cycling duration: initial hold step at 95 °C for 2 min, denaturation at 95 °C for 5 s, annealing/extension for 40 cycles at 60 °C for 30 s, and 60 °C for 5-s melting curve using Bio-Rad CFX96 Real-Time System (Hercules, CA, USA).

### 2.7. Real-Time Polymerase Chain Reaction (RT-PCR) with Individual Primers

RNA Extraction was completed the same way as described on RT-PCR Arrays. The cDNA strand was synthesized from the mRNA using iScript advanced reverse transcriptase kit from Bio-Rad. A solution of iScript advanced reaction mix (containing primers), reverse transcriptase, the sample, and water was added to clean 0.2 mL tubes. The thermal-cycling program included two steps: samples were first incubated and amplified following the manufacturer’s protocol (Bio-Rad). The sample (200 ng cDNA/reactions), master mix (Universal SYBR Green Supermix (item # 1725261)), primer, and nuclease-free water were mixed into each well. The unique assay ID for the Nrf2 primer was qMmuCID0021433, and for Keap1, qMmuCIP0028745, NF-kB1 qMmuCED0047222, NF-kB2 qMmuCED00450272, NF-kBIA qMmuCED0045043, Ikbkb qMuCID0005811, NIK qMmuCID0018436, and RELA qMmuCID0017564.

### 2.8. Capillary Western Analysis

Cells were treated with different treatments for 48 h. Treatments were as follows: control (cells + DMSO), cells treated with LPS (1 µg/mL) only or with HPT (100 µM) only, or a combination of HPT (100 µM) and LPS (1 µg/mL). Two days later, cells were collected, washed twice with PBS, and centrifuged to obtain the cell pellet. Cells were then sonicated with a protease inhibitor cocktail (total protein) and buffer. Protein concentration was measured using the bicinchoninic acid (BCA) Protein Assay Kit (Item # 23225). Standards from 0 to 2000 µg/mL concentration or samples and protein assay reagent were added to the 96-well plate. The Synergy H.T.X. Multi-Reader (BioTek: Winooski, VT, USA) was used to measure the concentration of proteins at 562 nm wavelength. The total protein expression was established by western capillary analysis.

Proteins were determined using automated Wes™ ProteinSimple (San Jose, CA, USA). ProteinSimple provided reagents, and the analysis was completed according to the manual. Optimization for each antibody and protein loading based on provided protocol was performed. Samples were mixed with a master mix for a final concentration of 0.2 mg/mL total protein, sample buffer, fluorescent molecular markers, and 40 mM dithiothreitol. Samples were then heated at 95 °C for 5 min. The dilution for samples was 1:5 to 1:25 and loaded into wells on the microplate and placed in the machine according to manufacturer guidelines. The reaction took place inside the machine. Target proteins were detected using a primary antibody, immunoadsorbed using a secondary antibody, and chemiluminescent substrates. Chemiluminescence captured by the device camera and digital image were analyzed and quantified using ProteinSimple Compass software (Version 6.2—San Jose, CA, USA).

### 2.9. Data Analysis

Statistical analysis was performed using GraphPad Prism (version 9.4.1) (San Diego, CA, USA). All data were expressed as the mean ± SEM from 3 independent experiments. Statistically significant differences in the experiments were calculated using a one-way ANOVA and Dunnett’s multiple comparison tests (* *p* < 0.05, ** *p* < 0.01, *** *p* < 0.001, **** *p* < 0.0001, and ns = non-signifigant). Gene expression was investigated using the CFX 3.1 Manager software (Bio-Rad, Hercules, CA, USA) and protein expression using ProteinSimple Compass software (Version 6.2—San Jose, CA, 95134, USA).

## 3. Results

### 3.1. Effect of Hesperetin and LPS on Cell Viability

Cell viability was assessed by using Alamar Blue^®^ reagent after 24 or 48 h. BV-2 microglial cells were treated with HPT only or co-treated with HPT and LPS. HPT treatment showed no statistically significant decrease in cell viability from 0.78 µM to 100 µM compared with the control. Still, as the concentration of HPT increased to 200 µM, there was a 20% decrease in viable cells in the 24 h treatment period (Figure 2A). When cells were treated with HPT 1 h before LPS, concentrations from 0.78 μM to 100 μM did not affect the cell viability of the cells. Still, the concentration of 200 μM showed 50% cytotoxicity (Figure 2B). At 48 h, the concentrations of HPT ranging from 0.78 μM to 200 μM showed that about 80% of cells were viable (Figure 2C). Results show that the HPT and LPS combination is less toxic than the control in 24 and 48 h treatments. Supported by the literature data obtained from cell viability assays, we used 100 μM of HPT and 1 μg/mL of LPS as our working concentrations for further evaluation.

### 3.2. Hesperetin’s Effect on Nitric Oxide Production

Results also showed that BV-2 cells treated with LPS presented a statistically significant increase in nitrate production compared with the control. When the cells were pretreated with HPT in concentrations from 0.78 μM to 12.5 μM and then activated with LPS, there was no statistically significant decrease in nitrite production compared with LPS alone. However, in concentrations ranging from 25 μM to 100 μM, HPT caused a significant decrease in nitrite production (Figure 3).

### 3.3. Hesperetin’s Effect on Antioxidant Mediators

The effect of HPT on the expression of CAT, SOD, and GSH was investigated in this study. Treatment with HPT-only or LPS-only did not show any statistically significant difference in CAT activity at 24 h compared with the control. However, when HPT was added 1 h before LPS, it inhibited the effects of LPS, decreasing the expression of CAT activity by 4.3-fold (Figure 4A). In 48 h treatment, HPT maintained the same level of CAT activity observed in control. However, LPS induced a statistically significant decrease, which was reverted by the combination of HPT and LPS, showing a 1.6-fold increase in CAT activity (Figure 4B).

The effect of HPT on SOD activity in BV-2 microglial cells at 24 h showed that the compound did not show any statistically significant modulation in SOD activity compared with the control. The same was observed when comparing LPS to control. However, when cells were treated with the combination of HPT and LPS, there was a decrease in SOD activity during the 24 h treatment period (Figure 5A). In the 48 h period, HPT treatment showed no statistically significant effect, and LPS significantly decreased SOD activity compared with the control. Contrary to the effect on the 24 h treatment, when cells were incubated with HPT 1 h before LPS, there was a significant increase in SOD activity compared with LPS, returning to control levels (Figure 5B).

The effect of HPT on GSH concentration in BV-2 microglial cells showed no statistically significant modulation compared with the control during the 24 h treatment period. However, there was a decrease in the GSH concentration in the LPS treatment compared with the control and a more significant reduction when the cells received the combination of HPT and LPS (Figure 6A). When comparing HPT to the control in the 48 h treatment, there was no statistically significant change in the GSH concentration. However, with the LPS treatment, there was a significant decrease in GSH levels. The co-treatment of HPT and LPS showed that HPT protected the cells, increasing GSH almost to the same level as the control, with a significant increase in the concentration of GSH compared with LPS alone (Figure 6B).

### 3.4. Hesperetin’s Effects on Oxidative Stress Gene Expression

RT-PCR arrays were used to profile the effect of HPT, LPS, and the co-treatment of HPT and LPS vs. the control on oxidative stress genes (Figure 7A). Results showed that treatment with LPS increased the mRNA expression of ERCC6 and NCF1. However, treatment with the HPT and LPS combination presented a statistically significant down-regulation of these genes, all known to be induced during oxidative stress (Figure 7B–D). Specifically, NOS2, which codes for inducible NOS, confirmed the results shown for NO production. PCR arrays also showed that treatment with LPS significantly decreased the expression of HMOX1 and GCLC. However, co-treatment with HPT and LPS upregulated the mRNA expression of these genes, which have antioxidant activity (Figure 8A,B).

### 3.5. Hesperetin Modulation of Keap1/Nrf2 Signaling

RT-PCR with individual primers for Nrf2 and Keap1 was performed. Data showed that in the LPS treatment, there was a decrease in the mRNA expression of Keap1 at 24 h compared with the control. At 48 h, no statistically significant difference between HPT and the control group was found. However, LPS treatment caused a significant decrease in Keap1 levels. In the co-treatment, both 24 and 48 h treatment periods showed that Keap1 expression was increased by 1.3 and 9-fold, respectively, compared with LPS only. This increase was much more accentuated at 48 h treatment, where HPT seems to reverse the effect of LPS (Figure 9A,B). Results also showed that HPT treatment presented a statistically significant increase in Nrf2 mRNA expression compared with the control at a 24 h treatment period. However, LPS did not cause any significant effect on Nrf2 mRNA expression compared with the control. The combination of HPT and LPS showed a 2.2-fold increase in Nrf2 expression compared with LPS. When HPT was incubated for 48 h, there was no statistical significance compared with the control. However, LPS treatment caused a significant decrease in Nrf2 expression. Moreover, after pretreatment with HPT for 1 h and then stimulation of cells with LPS, there was a 25-fold increase in Nrf2 mRNA expression at 48 h (Figure 9C,D). This increase in Nrf2 expression may be associated with the increased expression of antioxidants observed in this study.

### 3.6. Hesperetin’s Effect on Protein Expression of Keap1 and Nrf2

To investigate the effect of HPT on the protein expression of Keap1 and Nrf2 in BV-2 microglial cells, we performed capillary electrophoresis Wes analysis with specific antibodies against total Keap1 and Nrf2 proteins. The data demonstrated that activation with LPS did not significantly decrease the protein expression of Keap1. However, the combination of HPT and LPS significantly reduced protein expression over a 48 h treatment period (Figure 10A,B). The results also showed that LPS inhibited the expression of Nrf2 protein levels, but conversely, the combination of HPT and LPS induced a 25.7-fold increase compared with LPS treatment (Figure 11A,B). These results confirm the data on the transcription level, where Nrf2 expression was also increased when cells were pretreated with HPT.

### 3.7. Hesperetin’s Suppression of PD-L1 Gene

The effects of HPT on the gene expression of PD-L1 were investigated, and the data demonstrated that LPS significantly increased the PD-L1 mRNA expression by 25.6-fold compared with the control treatment. However, the combination of HPT and LPS decreased the expression of this gene by 50%, demonstrating that HPT protects BV-2 microglial cells against LPS effects (Figure 12).

### 3.8. Hesperetin’s Effects on NF-kB Signaling-Associated Genes and Proteins

HPT effects were evaluated on NF-kB-associated genes using individual RT-PCR primers. Data showed that NF-kB1 mRNA expression was increased 2.5-fold with the treatment of LPS compared with the control. However, the combination of HPT and LPS showed a 1.6-fold decrease in mRNA expression, suggesting HPT reverses the effects of LPS (Figure 13A). Results also demonstrated that on NF-kB2 expression, there was a 4.5-fold increase in the expression of this gene in the LPS treatment compared with the control, but when HPT was added 1 h before LPS, results showed a 1.8-fold decrease in the mRNA expression of NF-kB2 (Figure 13B). RT-PCR results also demonstrated that LPS treatment increased the mRNA expression of RELA by 3-fold compared with the control. HPT plus LPS showed a 3.3-fold decrease in RELA expression compared with the control (Figure 13C). Lastly, data showed that the mRNA expression of NIK was not increased in treatment with LPS compared with the control; however, there was a 4.3-fold decrease in a combination of HPT plus LPS compared with LPS (Figure 13D).

To confirm the results of RT-PCR at the protein level, capillary Wes analysis was used. Data showed that LPS induced protein expression for both molecular-weight proteins associated with NF-kB1. For NF-kB1/p50, there was a 4-fold increase in LPS treatment compared with the control. However, the combination of HPT and LPS showed a 3.4-fold decrease in p50 expression. The effect of HPT on NF-kB1/p105 showed a 4-fold reduction in protein expression when comparing the combination of HPT plus LPS and LPS treatment only (Figure 14A). The results also showed that LPS increased NF-kB2/p52 protein expression 3-fold compared with the control. However, the combination showed that HPT downregulated protein expression by 2-fold. The data showed that LPS induced NF-kB2/p100 protein expression by 3-fold compared with the control, but HPT, in combination with LPS, downregulated expression by 2.5-fold (Figure 14B). Results for RELA protein expression demonstrated an 8-fold increase in expression levels comparing LPS to control. The combination of HPT and LPS showed a 3-fold decrease compared with LPS (Figure 14C). We also investigated the HPT effect on phosphorylated NF-kB2 (phospho-p100) expression. Results showed that LPS increased its expression by 2.5-fold compared with the control, but the combination decreased the expression of the phosphorylated protein by 2.3-fold, almost reversing the effects of LPS (Figure 14D). These results further confirm the results obtained at the transcription level.

## 4. Discussion

Flavonoids have recently received considerable attention for their antioxidant and anti-inflammatory activities. Flavonoids are known for their ability to scavenge ROS, free radicals, and chelate metal ions. Flavonoids are vast and have been shown to modulate diseases such as diabetes, cardiovascular disease, and many types of cancer [1,2,3]. It has been indicated that multiple flavonoids can effectively inhibit neurotoxic pathways that are associated with neurodegenerative diseases. These compounds can be found in the daily diet because of their presence in many foods and beverages. Several studies have demonstrated that HPT exerts neuroprotective effects by neutralizing free radicals generated during cellular metabolism [39]. Also, HPT has been suggested to protect neurons against induced oxidative stress and inflammation in both in vivo and in vitro neurodegeneration models [3,17,40,41,42]. However, no data describes the antioxidative stress effects of HPT on BV-2 microglial cells.

The current study shows pretreatment with HPT offers cytoprotective effects and combats oxidative stress. This is confirmed by the antioxidant mediator studies. Specifically, the SOD enzyme catalyzes superoxide anion radicals into molecular hydrogen or hydrogen peroxide [43]. Enzymes work together to reduce elevated ROS and sustain healthy levels of oxidative stress [44]. CAT breaks down hydrogen peroxide into water, and oxygen is ubiquitous in most cell types [45]. The present study also investigated HPT’s effect on GSH concentration. GSH is most abundant in the CNS and acts as an important antioxidant in neuronal cells. Its deficiency has been suggested in aging and multiple diseases, including A.D [46].

Data from oxidative stress RT-PCR arrays confirmed that HPT modulates the expression of specific oxidative stress genes. Results show a significant decrease in *ERCC6* and *NCF1* mRNA expression with the co-treatment of HPT and LPS. These genes are overexpressed in oxidative stress and enhance tau phosphorylation, a critical factor in the formation of neurofibrillary tangles, which may be significant in the pathogenesis of A.D. [47]. PCR results also demonstrated that HPT decreased *NOS2* mRNA expression, which confirms the results showing a decrease in NO production. Reducing NO production may contribute to a reduction in inflammation and oxidative stress levels. Furthermore, HPT increased the gene expression of both *HMOX1* and *GCLC*. Upregulation of HMOX1 has been proven to exert neuroprotection due to the activation of Nrf2 [48]. This gene also exerts protection against oxidative injury and modulates inflammation [49,50]. Previously reported results indicated that HPT could activate HMOX1/Nrf2 signaling, which reduces oxidative stress by inducing the transcription of antioxidant genes [51]. The present study showed that PCR array results demonstrate that HPT may modulate the HMOX1 pathway, reducing oxidative stress. Inducing the expression of this gene coupled with Nrf2 activation may also be critical in slowing the progression of A.D. [16,20,52]. Nrf2 is ubiquitously expressed, specifically in the brain, and it is a crucial defense mechanism against glial cell and neuron toxicity [15]. This transcription factor upregulates several antioxidant enzymes, increases anti-inflammatory mediator expression, and initiates mitochondrial signaling pathways [16]. Nrf2 is important in the crosstalk between antioxidant and anti-inflammatory pathways, which are said to be secondary effects of Nrf2 antioxidant activity [17]. Nrf2 is normally bound to Kelch-like E.C.H.-Associated Protein 1 (Keap1), which targets it and causes it to degrade via the proteasomes [8,18,19,20,21,22]. In the presence of oxidative stress, Keap1 functional groups are modified, resulting in conformational changes that reduce Keap1 and degrade Nrf2, which allows it to accumulate in the cell’s nucleus [15]. RT-PCR arrays also showed that HPT increased the expression of *GCLC*, the rate-limiting enzyme for extracellular glutathione synthesis, which would also be beneficial in fighting oxidative stress [46]. It has been suggested that GCLC activity reduces with age, which elevates ROS levels [46]. The data from this study indicate that HPT could potentially decrease ROS production, which is a key factor in oxidative stress.

Furthermore, the data show that the co-treatment of HPT and LPS increased *Keap1* mRNA expression with opposite effects on the protein level, suggesting that Keap1 may go through posttranslational modifications. The results also show that HPT significantly upregulated *Nrf2* gene expression by 25-fold, even when cells were stimulated by LPS, which was confirmed at the protein level by a 25.7-fold increase. This significant upregulation of Nrf2 levels may be associated with the increased production of antioxidant genes that was observed in this study. As previously stated, Keap1 and Nrf2 are bound in a complex, and normally ubiquitin-proteosomes degrade Nrf2 to maintain homeostatic conditions [9,53,54]. However, chronic ROS dissociates Nrf2 from Keap1, which translocates into the nucleus of cells, where Nrf2 induces the transcription of several antioxidant genes [15,55], some of which have been discussed here. It has also been demonstrated that Nrf2 prevents the accumulation of tau proteins through the enhancement of autophagic degradation [56]. The results of this study demonstrate that HPT might offer neuroprotection through the activation of Nrf2 signaling, which may be linked to the increased expression of antioxidants.

The present study also demonstrates that HPT reduced the gene expression of PD-L1, which has been reported to be upregulated as individuals age. An increased expression of this gene is observed in chronic inflammation and is associated with A.D. development and progression [29]. As previously stated, it has been reported that PD-1 is expressed in microglial cells, and the PD-1/PD-L1 pathway was demonstrated to be upregulated by LPS [57]. The present study reports that HPT reduced the expression of PD-L1, which could ameliorate chronic neuroinflammation in neurodegenerative diseases, specifically A.D.

Furthermore, the data showed that multiple genes associated with NF-kB signaling were modulated by HPT treatment. This family of transcription factors has been shown to have various duties in cellular processes, but more specifically in inflammation. Induction of this pathway promotes the transcription of pro-inflammatory mediators. Its activation is via canonical and non-canonical pathways, which have been extensively studied. Activation of the canonical pathway initiates the release of dimers p65/p50 in the cytoplasm coupled to IkBα. These dimers then translocate into the nucleus of the cell and bind to DNA, leading to the expression of pro-inflammatory cytokines. Additionally, activating the non-canonical pathway induces the phosphorylation of Ikkα by NIK, leading to the phosphorylation of p100 and then generating p52 [58]. Data showed that HPT decreased the expression of NF-kB1, NF-kB2, and RELA at both the transcriptional and protein levels. Also, HPT inhibited the protein expression of phospho-NF-kB2. The data indicate that HPT may regulate the production of various pro-inflammatory cytokines by modulating NF-kB signaling-associated proteins.

Nrf2 and NF-kB are critical mediators in response to chronic oxidative stress and inflammation. The literature demonstrates that the downregulation of Nrf2 increases the expression of NF-kB, which increases the production of pro-inflammatory cytokines in mice, suggesting that a decrease or loss in Nrf2 could enhance inflammation more aggressively [59]. Although regulating these pathways is critical for chronic inflammation and oxidative stress, the mechanism associated with their crosstalk has not been elucidated. Therefore, this study indicates the potential of HPT to reduce oxidative stress through the upregulation of Nrf2 and to inhibit chronic inflammation via the downregulation of NF-kB signaling, confirming the literature that shows that one pathway directly or indirectly affects the other.

In this investigation, LPS-stimulated BV-2 microglial cells were used. This is a well-characterized model used in research, especially in the study of neurodegenerative disorders involving immune responses, including oxidative stress and neuroinflammation. Many articles describe that BV-2 cells are compatible substitutes for the use of primary microglia in several experimental assays as well as in studies of a more complex nature involving cell–cell interaction [60]. Studies described that comparing BV-2 cell lines with primary rat microglia showed that LPS stimulation caused BV-2 cells to secrete lesser but still significant amounts of NO compared with primary microglia [61]. Henn et al. [60] investigated the BV-2 cells as a suitable alternative to the primary cultures. They described that in response to LPS, 90% of genes induced in the BV-2 cells were also induced in primary microglia, indicating that this is a good research model. However, further studies will be needed using in vivo models to validate and elucidate the molecular mechanisms used by hesperetin to ameliorate oxidative stress and inflammation.

## 5. Conclusions

The current investigation showed that HPT reduces oxidative stress and inflammation by downregulating the expression of inflammatory and oxidative stress mediators. HPT downregulated the production of NO and levels of *NOS2* mRNA expression, which can aid in reducing both inflammation and ROS. The compound could also decrease oxidative damage by increasing GSH concentration and CAT and SOD activities. Furthermore, HPT reduced the mRNA expression of *ERCC6* and *NCF1*, known to increase levels of oxidative stress, and increased the mRNA expression of the antioxidant genes *HMOX1* and *GCLC*. Additionally, HPT increased the expression of Nrf2 and modulated Keap1 expression at the transcriptional and protein levels, suggesting that its antioxidant effects may be exerted via the Keap1/Nrf2 signaling pathway. This compound also significantly modulated the expression of NF-kB-associated genes and proteins and inhibited PD-L1 gene expression, demonstrating that HPT can reduce inflammation, which is also a critical factor in the progression of A.D. Moreover, aging also plays an important role in neurodegenerative diseases, specifically A.D. As one ages, the levels of antioxidants decrease, resulting in the overproduction of ROS. In this investigation, HPT has shown the potential to reduce chronic inflammation and oxidative stress and protect the BV-2 microglial cells. These findings indicate that HPT may be useful in preventing the onset or slowing the progression of neurodegenerative disorders associated with excessive oxidative stress and neuroinflammation in the brain. HPT’s ability to alter A.D. mechanistically may be critical in treating this disease with few adverse effects. In conclusion, the findings indicate that HPT may be a potential candidate for neurodegenerative therapy targeting the activation of Nrf2 and the inhibition of NF-kB-associated genes, which may help slow A.D. onset and progression (Figure 15).

## Figures and Tables

**Figure 1 brainsci-13-01144-f001:**
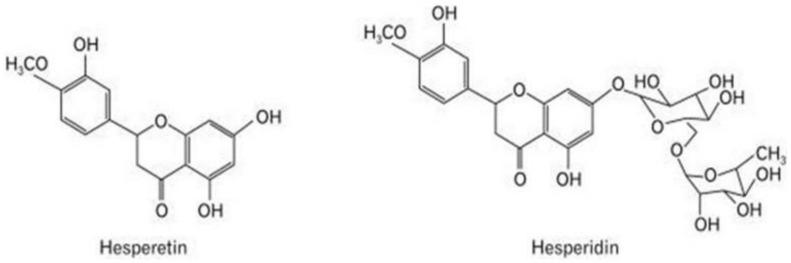
Chemical structure of hesperetin and naturally occurring hesperidin [17].

**Figure 2 brainsci-13-01144-f002:**
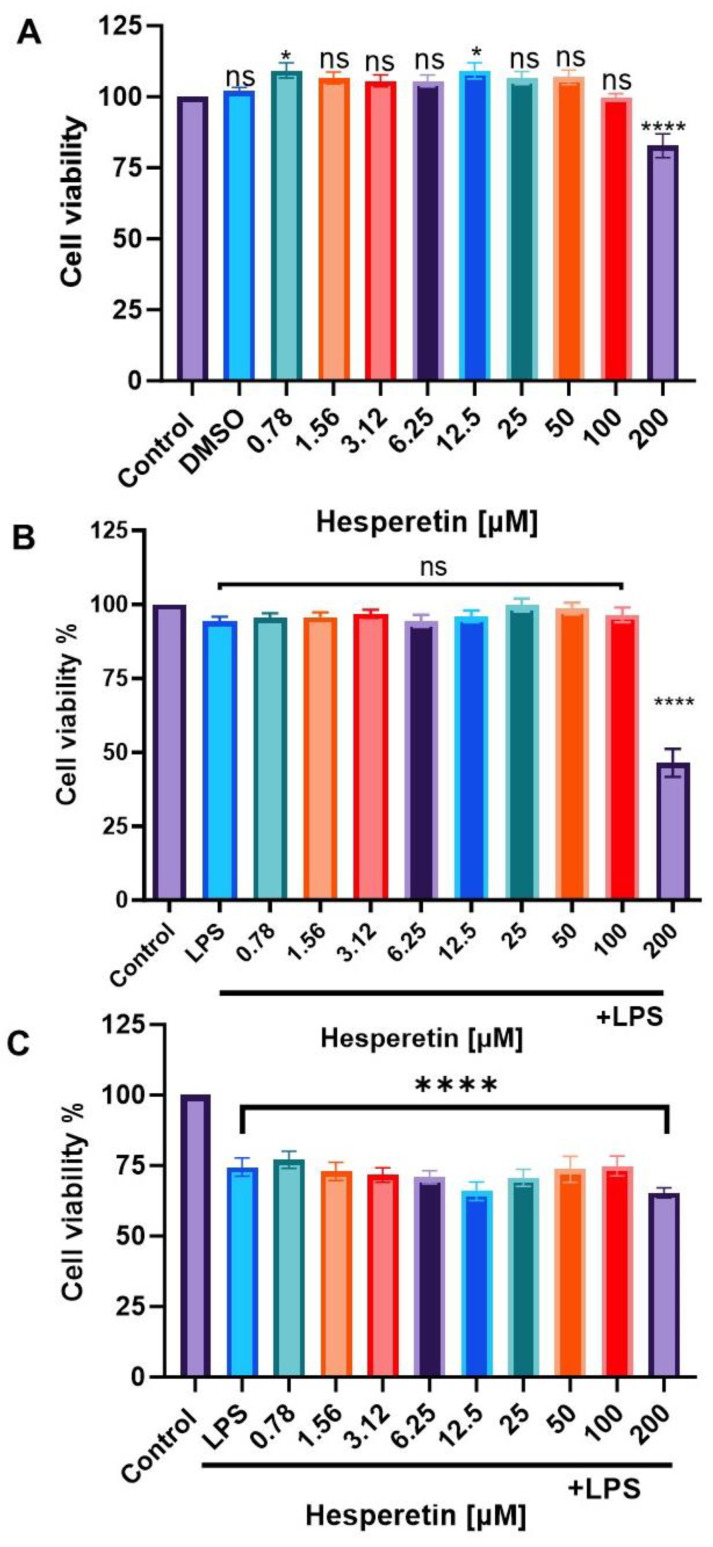
(**A**) the effect of hesperetin treatment (0.78–200 µM) on cell viability of BV-2 microglial cells for 24 h; (**B**) The effect of the combination of hesperetin and LPS (1 μg/mL) on cell viability of BV-2 microglial cells for 24 h; (**C**) The effect of hesperetin and LPS on cell viability of BV-2 microglial cells for 48 h. All the experiments were performed at 5% CO_2_/atm for 24 or 48 h. The data are presented as mean ± SEM (3 independent experiments), and the significance of differences from the control was determined by a one-way ANOVA and Dunnett’s multiple comparison tests.* *p* < 0.005, **** *p* < 0.0001, ns = non-signifigant.

**Figure 3 brainsci-13-01144-f003:**
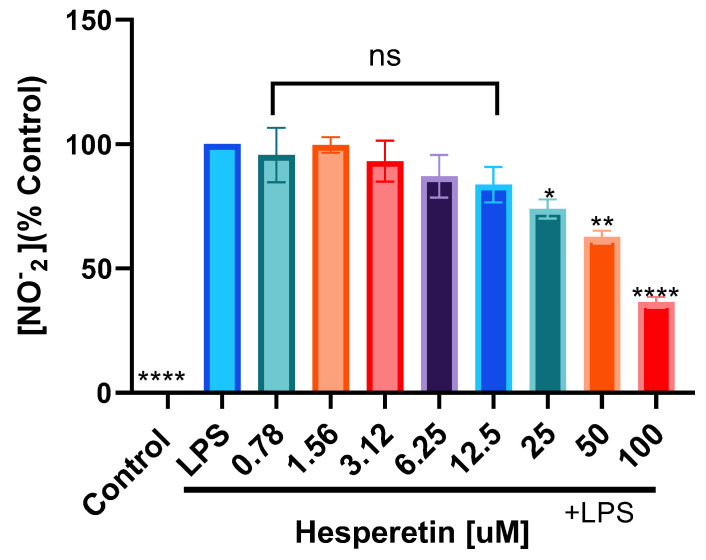
The effect of hesperetin on nitrite production (%) on LPS-activated BV-2 microglial cells stimulated with LPS (1 μg/mL). The data are presented as mean ± SEM (3 independent experiments). A one-way ANOVA evaluated the significance of the difference between LPS and treatments, followed by Dunnett’s multiple comparison test * *p* < 0.05, ** *p* < 0.01, **** *p* < 0.0001, ns = nonsignificant.

**Figure 4 brainsci-13-01144-f004:**
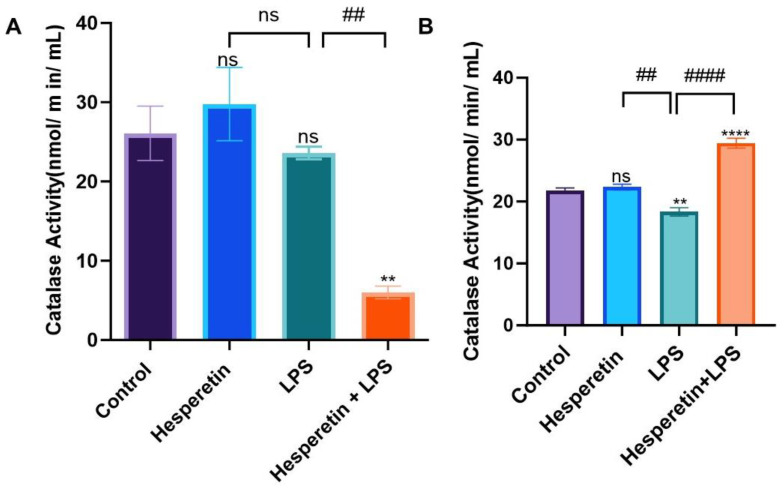
The effect of hesperetin on catalase activity with respective treatments at 24 h (**A**) and 48 h (**B**). The data are presented as mean ± SEM (3 independent experiments). A one-way ANOVA evaluated the significance of the difference between control vs. all treatments (** *p* < 0.01 and **** *p* < 0.0001), and LPS vs. all treatments (## *p* < 0.01 and #### *p* < 0.0001), followed by Dunnett’s multiple comparison test. ns = non-significant.

**Figure 5 brainsci-13-01144-f005:**
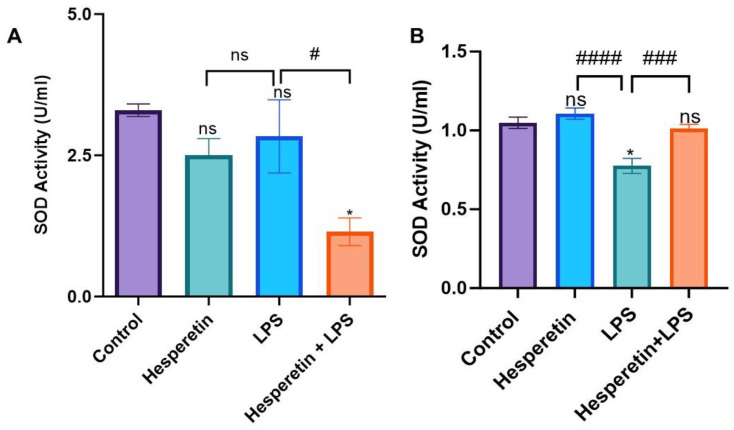
The effect of hesperetin on superoxide dismutase activity with respective treatments at 24 h (**A**) and 48 h (**B**). The data are presented as mean ± SEM (3 independent experiments). The significance of the difference between control vs. all treatments (* *p* < 0.05) and LPS vs. all treatments (# *p* < 0.05, ### *p* < 0.001, and #### *p* < 0.0001) was evaluated by a one-way ANOVA, followed by Dunnett’s multiple comparison test. ns = non-significant.

**Figure 6 brainsci-13-01144-f006:**
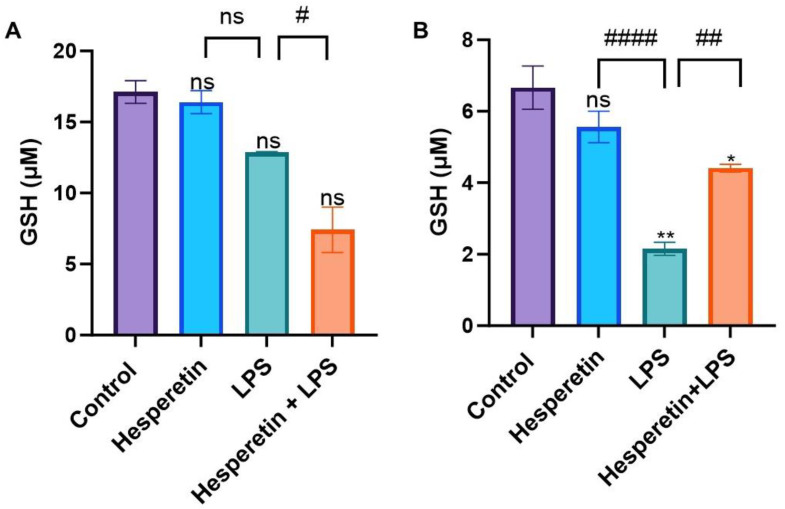
The effect of hesperetin on glutathione concentration (μM) with respective treatments at 24 h (**A**) and 48 h (**B**). The data are presented as mean ± SEM (3 independent experiments). A one-way ANOVA evaluated the significance of the difference from control vs. all treatments (* *p* < 0.05, ** *p* < 0.01) and LPS vs. co-treatment (# *p* < 0.05, ## *p* < 0.01, #### *p* < 0.0001), followed by Dunnett’s multiple comparison test. ns = non-significant.

**Figure 7 brainsci-13-01144-f007:**
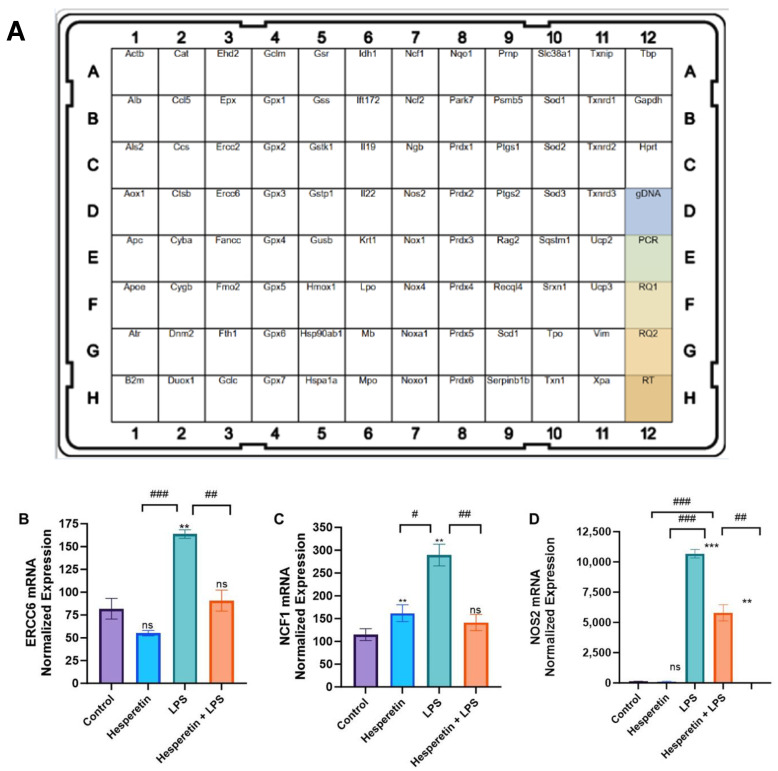
(**A**) PCR Array layout. Positive controls are in the lower-right column. (**B**) Effect of hesperetin on normalized mRNA expression of ERCC6, (**C**) NCF1, and (**D**) NOS_2_. The data are presented as mean ± SEM (3 independent experiments). A one-way ANOVA evaluated the significance of the difference between control vs. all treatments (** *p* < 0.01 and *** *p* < 0.001) and LPS vs. all treatments (# *p* < 0.05, ## *p* < 0.01, and ### *p* < 0.001), followed by Dunnett’s multiple comparison test. ns = non-significant.

**Figure 8 brainsci-13-01144-f008:**
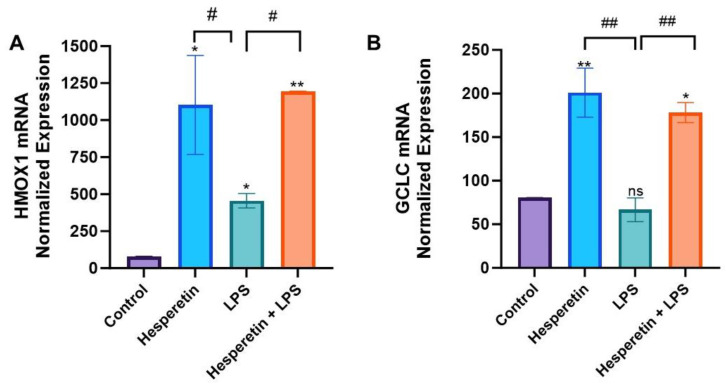
Effect of hesperetin on normalized mRNA expression of (**A**) HMOX1 and (**B**) GCLC. The data are presented as mean ± SEM (3 independent experiments). A one-way ANOVA evaluated the significance of the difference between control vs. all treatments, (* *p* < 0.05, ** *p* < 0.01) and LPS vs. all treatments (# *p* < 0.05, ## *p* < 0.01), followed by Dunnett’s multiple comparison test. ns = non-significant.

**Figure 9 brainsci-13-01144-f009:**
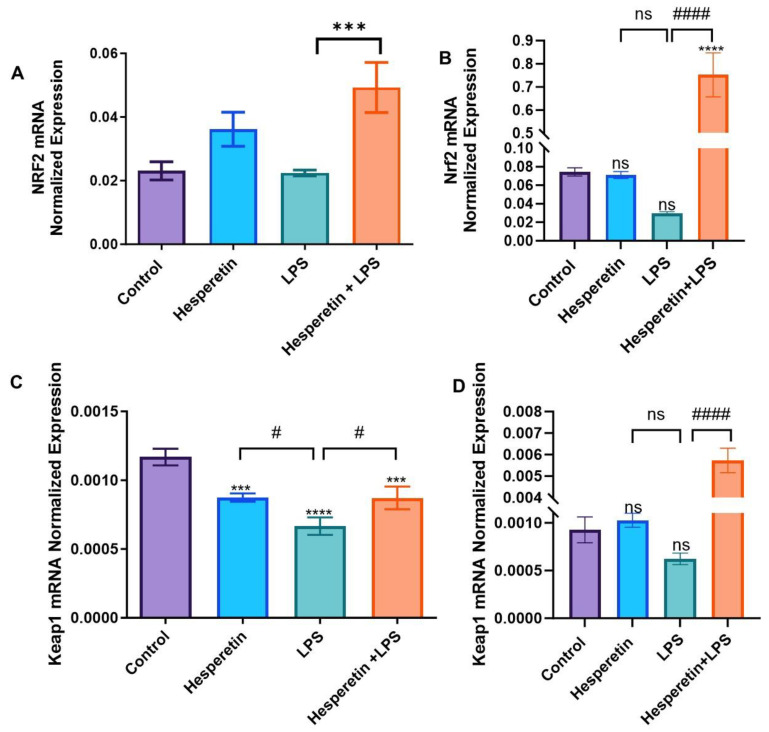
The effect of hesperetin on Keap1 (**A**) 24 h and (**B**) 48 h and Nrf2 mRNA normalized expression (**C**) 24 h and (**D**) 48 h. The data are presented as mean ± SEM (3 independent experiments). A one-way ANOVA evaluated the significance of the difference between control vs. all treatments (*** *p* < 0.001, and **** *p* < 0.0001) and LPS vs. all treatments (# *p* < 0.05 and #### *p* < 0.0001), followed by Dunnett’s multiple comparison test. ns = nonsignificant.

**Figure 10 brainsci-13-01144-f010:**
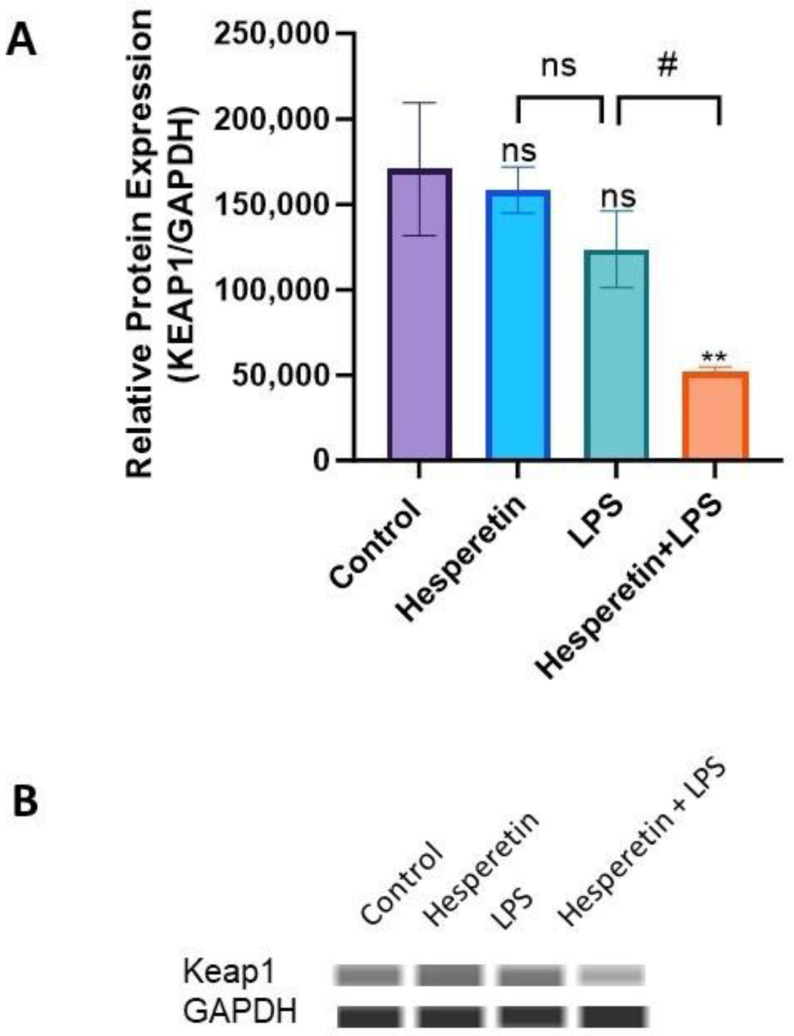
(**A**) The effect of hesperetin on Keap1 protein expression. (**B**) Bands representing the protein expression of four treatments at 48 h. The data are presented as mean ± SEM (3 independent experiments). A one-way ANOVA evaluated the significance of the difference between control vs. all treatments (** *p* < 0.010 and LPS vs. all treatments (# *p* < 0.05), followed by Dunnett’s multiple comparison test. ns = non-significant.

**Figure 11 brainsci-13-01144-f011:**
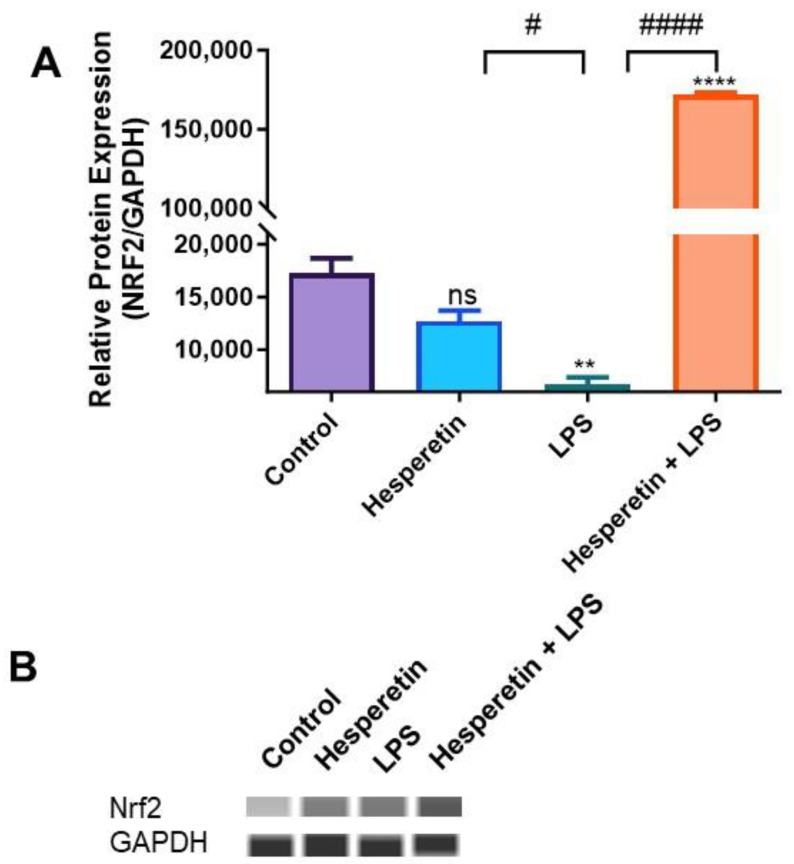
(**A**) The effect of hesperetin on Nrf2 protein expression. (**B**) Bands representing the protein expression after treatments at 48 h. The data are presented as mean ± SEM (3 independent experiments). A one-way ANOVA evaluated the significance of the difference between control vs. all treatments (** *p* < 0.01 and **** *p* < 0.0001) and LPS vs. all treatments (# *p* < 0.01 and #### *p* < 0.0001), followed by Dunnett’s multiple comparison test. ns = non-significant.

**Figure 12 brainsci-13-01144-f012:**
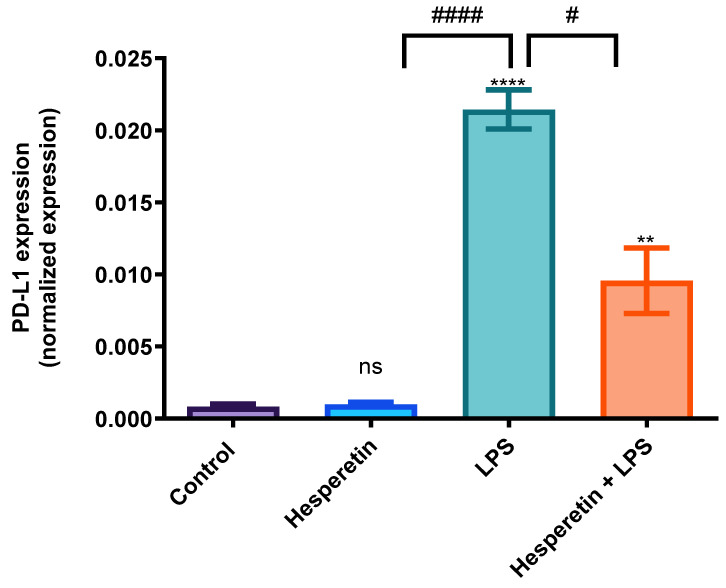
Hesperetin’s effect on PD-L1 mRNA normalized expression. The data are presented as mean ± SEM (3 independent experiments). A one-way ANOVA assessed the significance of the difference between control vs. treatments (** *p* < 0.01 and **** *p* < 0.0001) and LPS vs. treatments (# *p* < 0.05, #### *p* < 0.0001), followed by Dunnett’s multiple comparison test. ns = non-significant.

**Figure 13 brainsci-13-01144-f013:**
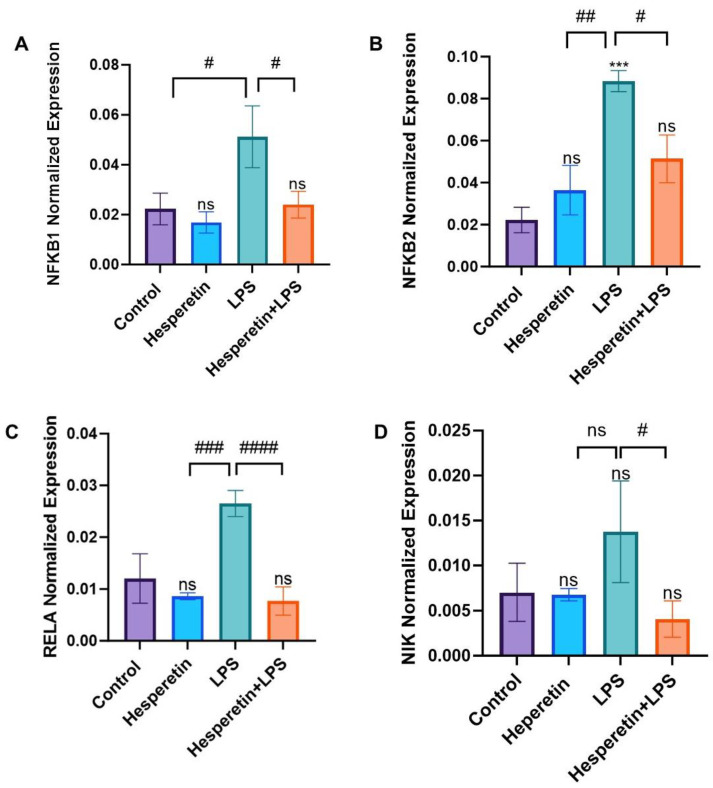
The effect of hesperetin on (**A**) NF-kB1, (**B**) NF-kB2, (**C**) RELA, and (**D**) NIK mRNA normalized gene expression. The data are presented as mean ± SEM (3 independent experiments). A one-way ANOVA evaluated the significance of the difference between control vs. all treatments (*** *p* < 0.001) and LPS vs. all treatments (# *p* < 0.05, ## *p* < 0.01, ### *p* < 0.001, and #### *p* < 0.0001), followed by Dunnett’s multiple comparison test. ns = non-significant.

**Figure 14 brainsci-13-01144-f014:**
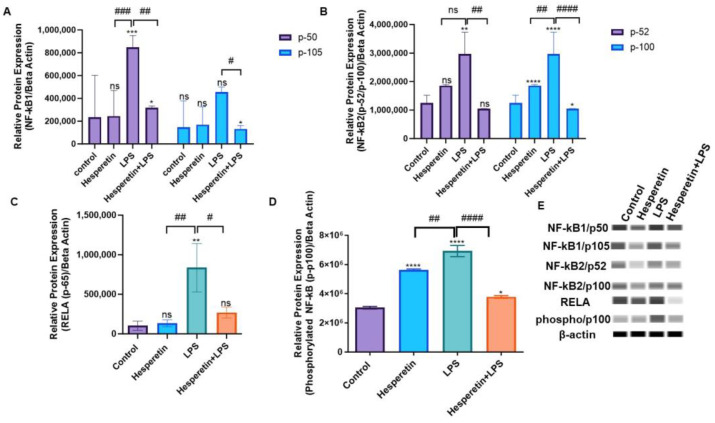
Hesperetin’s Effect on NF-kB-associated proteins. NF-kB1 (**A**), NF-kB2 (**B**), RELA (**C**), *p*-NF-kB2 (**D**). Bands represent the protein expression in each treatment. The data are presented as mean ± SEM (3 independent experiments). A one-way ANOVA assessed the significance of the difference between control vs. treatments (* *p* < 0.05, ** *p* < 0.01, *** *p* < 0.001, and **** *p* < 0.0001) and LPS vs. treatments (# *p* < 0.05, ## *p* < 0.01, ### *p* < 0.001, #### *p* < 0.0001), followed by Dunnett’s multiple comparison test., ns = nonsignificant.

**Figure 15 brainsci-13-01144-f015:**
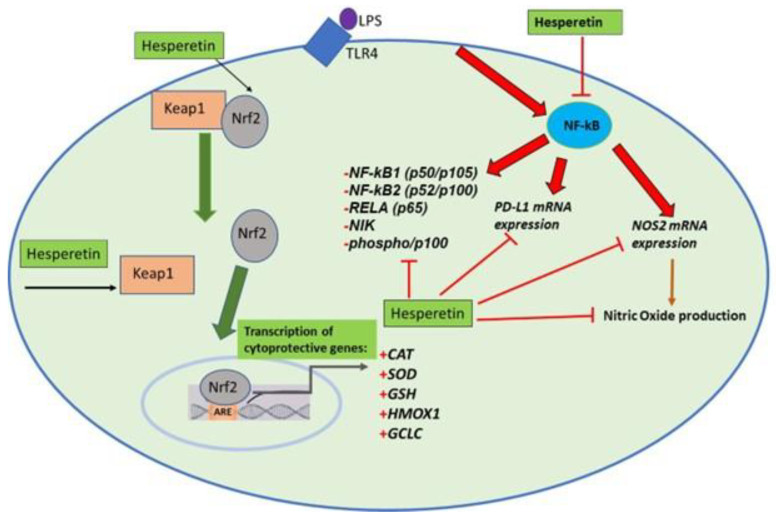
Proposed mechanism of hesperetin on LPS-activated BV-2 microglial cells. The figure shows the effect of hesperetin on Nrf2/Keap1 and NF-kB signaling.

## Data Availability

All the data is included within the text.

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
