# Peer review of "Involvement of Nrf2 Activation and NF-kB Pathway Inhibition in the Antioxidant and Anti-Inflammatory Effects of Hesperetin in Activated BV-2 Microglial Cells"

_brainsci, 2023, doi:10.3390/brainsci13081144_

Round 1
Reviewer 1 Report
Comments and Suggestions for Authors
In the submitted manuscript Evans et. al. investigated the antioxidant and anti-inflammatory effects of flavonoid hesperetin (HPT) on LPS-activated murine BV-2 microglial cells, a well-established model to investigate neurodegeneration. The molecular mechanisms of HPT in the modulation of oxidative stress through the Nrf2/Keap1 signaling pathway, as well as its effects on the inflammatory process through the regulation of NF-kB and PD-1/PD-L1 signaling pathways, were studied.
The submitted manuscript is interesting and well-written. However, the following comments should be addressed:
Section Materials and Methods
References to the applied protocols should be provided in all subsections.
Section Results
Figure 7: The font size in Figure 4 A should be increased to be clearly visible.
The interpretation of the results in lines 448-450 is unclear and should be revised.
Figure 14: The font size on the y- and the x-axis of Figures A-D should be increased to be clearly visible.
Section Discussion
Line 482: Known beneficial biological effects of flavonoids should be briefly described. The beneficial biological effects of flavonoids were comprehensively reviewed in recent articles.
References:
1. Brglez Mojzer, E.; Knez Hrnčič, M.; Škerget, M.; Knez, Ž.; Bren, U. Polyphenols: Extraction Methods, Antioxidative Action, Bioavailability and Anticarcinogenic Effects. Molecules 2016, 21, 901. https://doi.org/10.3390/molecules21070901
2. Furlan, V.; Bren, U. Helichrysum italicum: From Extraction, Distillation, and Encapsulation Techniques to Beneficial Health Effects. Foods 2023, 12, 802. https://doi.org/10.3390/foods12040802
Lines 487-509: The detailed interpretation of the results was already provided in the section Results and does not need to be repeated.
Lines 581-582: The insights into inhibitory mechanisms of flavonoid HPS on antioxidant proteins as well as on inflammatory proteins involved in Alzheimer's disease progression can be revealed through advanced molecular dynamics techniques and free-energy calculations as well as through inverse molecular docking.
References:
1. Pantiora, P.; Furlan, V.; Matiadis, D.; Mavroidi, B.; Perperopoulou, F.; Papageorgiou, A.C.; Sagnou, M.; Bren, U.; Pelecanou, M.; Labrou, N.E. Monocarbonyl Curcumin Analogues as Potent Inhibitors against Human Glutathione Transferase P1-1. Antioxidants 2023, 12, 63. https://doi.org/10.3390/antiox12010063
2. Furlan, V.; Bren, U. Insight into Inhibitory Mechanism of PDE4D by Dietary Polyphenols Using Molecular Dynamics Simulations and Free Energy Calculations. Biomolecules 2021, 11, 479. https://doi.org/10.3390/biom11030479
3. Kores, K.; Kolenc, Z.; Furlan, V.; Bren, U. Inverse Molecular Docking Elucidating the Anticarcinogenic Potential of the Hop Natural Product Xanthohumol and Its Metabolites. Foods 2022, 11, 1253. https://doi.org/10.3390/foods11091253
Author Response
We are pleased to resubmit the revised Manuscript ID: brainsci-2524030. Title: "Involvement of Nrf2 Activation and NF-kB Pathway Inhibition in the Antioxidant and Anti-inflammatory Effects of Hesperetin in Activated BV-2 Microglial Cells". We appreciate the reviewers' constructive criticisms and comments and have addressed each of their concerns as outlined below.
Reviewer 1
- Section Materials and Methods
References to the applied protocols should be provided in all subsections.
Response: Cell viability, Nitric Oxide, and PCR protocols were optimized in our lab and followed as described in the material and methods. Glutathione, catalase, and superoxide dismutase assays were performed according to the manufacturer's protocol. The catalog number for each assay was added to "4. Materials and Methods". PCR array assays followed Bio-Rad protocol and ProteinSimple Western Analysis according to ProteinSimple protocol. The catalog number for both of these assays was also included to section "4. Materials and Methods".
- Section Results
Figure 7: The font size in Figure 4 A should be increased to be visible. The interpretation of the results in lines 448 - 450 is unclear and should be revised. Figure 14: The font size on the y- and the x-axis of Figure A-D should be increased to be visible.
Response: We appreciate the reviewer's comments. All graphs have been revised to be visible, and the font size increased, as requested. The description of results was revised (last sentence of: 3.8. Hesperetin's Effects on NF-kB Signaling Associated Genes and Proteins).
- Section Discussion
Line 482: Known beneficial biological effects of flavonoids should be briefly described. The beneficial biological effects of flavonoids were comprehensively reviewed in recent articles.
Response: We appreciate the reviewer's suggestion, and we added information on the beneficial biological effects of flavonoids to the Discussion section (paragraph 1; lines 2 - 7).
Lines 487-509: The detailed interpretation of the results was already provided in the section Results and did not need to be repeated.
Response: We agreed with the reviewer's comments, and lines 487-509 were modified (Discussion section: paragraph 2).
Lines 581-582: The insights into inhibitory mechanisms of flavonoid HPS on antioxidant proteins and on inflammatory proteins involved in Alzheimer's disease progression can be revealed through advanced molecular dynamics techniques and free-energy calculations as well as through inverse molecular docking.
Response: We appreciate the reviewer's comments and the suggested references. It is known that several research groups have been using molecular docking focused on structure-based virtual screening for the identification of new compounds towards a specific target protein. Although the present study focused on the signaling pathway involved in hesperetin's anti-inflammatory and antioxidant effects, we will consider the use of advanced molecular dynamics techniques, free-energy calculations, and inverse molecular docking in our future studies.
Reviewer 2 Report
Comments and Suggestions for Authors
This is an interesting manuscript where the anti-oxidative and anti-inflammatory effects of flavonoids are investigated in a known cell model of neurodegeneration. The results show that genes involved in inflammation and oxidative stress are modulated in response to Hesperitin treatment. While the premise of the manuscript is overall convincing, there are several minor issues that need to be addressed.
1. A little bit more introduction on Hesperitin is required in the introduction section. What is the physiological concentration of Hesperitin that has a beneficial activity in the body? Is 100 um that was chosen by the authors a comparable dose?
2. What was the power of each experiment? the power needs to be included in the figure legend near the p-value. In several experiments (catalase activity, ), the effect of LPS and Hesperitin contradict at different time points. This was not well explained in the discussion. Are the contradicting effects a result of insufficient power?
3. What is the effect of LPS on the various genes investigated here? The effect of LPS does not seem to be very consistent. A literature search needs to be performed on what genes are significantly impacted by LPS in BV-2 mouse microglial cells. It is not clear if the authors performed a preliminary experiment to check if LPS is indeed causing inflammatory/oxidative stress in these cells.
4. The labels in the figures are confusing. Please update the figures to be clearly understood. Also, include both tails of the error bars indicating standard deviation. Failure to do so complicates the interpretation of certain figures. eg. Fig 8A: the authors claim the normalized expression of HMOX1 mRNA in Hesperitin-only and LPS-only treated cells is not significantly different. But the differences in the bars suggest otherwise. Only the upper tail of standard deviation error bar is indicated here. Similarly, the authors claim there is a significant difference between the cells treated with LPS only and Hesperitin + LPS, but there is no error bar for that treatment; and Hespertin-only treatment and LSP+hesperitin treatments show expression at the same level. Hence, the figures need a thorough revision.
5. What is the rationale behind including 24 hr treatment time point? It seems to me that most targets that are being investigated are showing the expected trend at 48 hrs. Maybe the 24-hour time point treatments can be included as a supplemental figure?
6. The authors have not explained their rationale behind choosing BV-2 murine cell line for these experiments. there are other human microglial cell lines available (eg. HMC3 cells) that would have given far more clinically relevant results. Do the authors intend to perform in vivo mouse studies to further investigate the effects of Hesperitin in preventing neurodegeneration? Ultimately unless going towards validating these findings in an animal model, the use of this cell line may be a major limitation for this manuscript. Is any information available on the relevance or similarity between the secretome of BV-2 murine microglial cells and in vivo human microglial cells?
7. Limitations section is missing in the manuscript. Although, the discussion section covers the significance of the findings of this paper, a limitations section is necessary to list and discuss the shortcomings of this paper as well as its limited scope.
Author Response
We are pleased to resubmit the revised Manuscript ID: brainsci-2524030. Title: "Involvement of Nrf2 Activation and NF-kB Pathway Inhibition in the Antioxidant and Anti-inflammatory Effects of Hesperetin in Activated BV-2 Microglial Cells". We appreciate the reviewers' constructive criticisms and comments and have addressed each of their concerns as outlined below.
Reviewer 2
- A little more introduction on Hesperitin is required in the introduction section. What is the physiological concentration of Hesperitin that has a beneficial activity in the body? Is 100 um that was chosen by the authors a comparable dose?
Response: Hesperetin is the aglycone metabolite from hesperidin, and currently, hesperidin is sold as a supplement in healthy stores. It is available for purchase at the dose of 500 mg twice daily, indicated for circulation and heart health. Hesperetin is also called the active form of hesperidin, which, unlike its aglycone, has poor membrane permeability and hereafter is mainly absorbed through the paracellular pathway, which means that the tight link of intestinal cells may limit its absorption. Considering this limitation, several papers describe different drug delivery methods, such as solid lipid nanoparticles loaded with hesperidin and hesperidin-β-CD inclusion complexes (1).
On the other hand, studies indicate that hesperetin can cross the blood-brain barrier, which is a significant factor considering its neuroprotective effects. This compound can reach the disease site in the case of neurodegenerative disorders, acting directly where the pathological process is occurring (2).
In the present study, 100µM was the concentration chosen based on the cell viability results showing that about 80% of the cells were alive after combining hesperetin and LPS. However, this concentration will have to be tested in our future in vivo studies to obtain a comparable dose that would have beneficial activity in the body.
As requested, more information on hesperetin was included in the introduction section (paragraph 5; lines (11 – 16).
- Wdowiak K, Walkowiak J, Pietrzak R, Bazan-Woźniak A, Cielecka-Piontek J. Bioavailability of Hesperidin and Its Aglycone Hesperetin-Compounds Found in Citrus Fruits as a Parameter Conditioning the Pro-Health Potential (Neuroprotective and Antidiabetic Activity)-Mini-Review. Nutrients. 2022 Jun 26;14(13):2647. doi: 10.3390/nu14132647. PMID: 35807828; PMCID: PMC9268531.
- Makarova NM. [Bioavailability and metabolism of flavonoids]. Eksp Klin Farmakol. 2011;74(6):33-40. Russian. PMID: 21870774.
- What was the power of each experiment? The power needs to be included in the figure legend near the p-value. In several experiments (catalase activity), the effect of LPS and Hesperitin contradict at different time points. This was not well explained in the Discussion. Are the contradicting effects a result of insufficient power?
Response: For each experiment, there were 3 biological replicates and at least 3 technical replicates to validate the results. As requested, this information was added to each one of the legends of the graphs. We agree that the effects of LPS and hesperetin are contradictory at different time points. At the beginning of our investigation, we had 24-h studies as our goal to investigate hesperetin effects on the LPS-stimulated BV-2 microglial cells. However, we observed a reduction of the antioxidant activity at 24h. As we increased the period of incubation to 48h, we observed opposite effects, showing that hesperetin may need more time to exert its antioxidant effect. These data are consistent among the antioxidant genes and proteins studied and with the expression of Nrf2, where we see a higher modulation after a 48-h period. Therefore, the contradiction is not due to insufficient power, but it is part of the molecular mechanism of the compound.
- What is the effect of LPS on the various genes investigated here? The effect of LPS does not seem to be very consistent. A literature search needs to be performed on what genes are significantly impacted by LPS in BV-2 mouse microglial cells. It is not clear if the authors performed a preliminary experiment to check if LPS is indeed causing inflammatory/oxidative stress in these cells.
Response: We appreciate the reviewer's comments. Lipopolysaccharide (LPS) is present in gram-negative bacteria, and in the present study, LPS was used to induce a biological inflammatory response in the BV-2 microglial cells. These cells are known for their function of clearing debris in the brain. Specifically, in AD, an accumulation of amyloid beta plaques causes the microglia to be over-activated, initiating chronic inflammation and oxidative stress.
In this investigation, we used RT-PCR arrays to screen 91 genes associated with oxidative stress. There is no way to predict the specific genes that will be modulated by each one of the treatments (control, hesperetin, LPS, and combination of hesperetin + LPS), but the normalized expression of the target genes showed us the differences between the effect of each one of these treatments. The results are consistent considering that LPS induces inflammation and oxidative stress, and the data describes that LPS downregulated some antioxidant genes, such as HMOX1 and GCLC, and upregulated oxidative stress genes, such as ERCC6, NCF1, and NOS2.
The use of LPS to stimulate BV-2 cells is a well-established model to investigate compounds' anti-inflammatory and antioxidant properties. Our laboratory has been using this model for many years, and the literature presents many papers using the same model. Please see below some of the references.
- Barber, K.; Mendonca, P.; Evans, J.A.; Soliman, K.F.A. Antioxidant and Anti-Inflammatory Mechanisms of Cardamonin through Nrf2 Activation and NF-kB Suppression in LPS-Activated BV-2 Microglial Cells. Int. J. Mol. Sci. 2023, 24, 10872. https://doi.org/10.3390/ijms241310872
- Mendonca P, Taka E, Bauer D, Reams RR, Soliman KFA. The attenuating effects of 1,2,3,4,6 penta-O-galloyl-β-d-glucose on pro-inflammatory responses of LPS/IFNγ-activated BV-2 microglial cells through NFƙB and MAPK signaling pathways. J Neuroimmunol. 2018 Nov 15;324:43-53. doi: 10.1016/j.jneuroim.2018.09.004. Epub 2018 Sep 11. PMID: 30236786; PMCID: PMC6245951.
- Horvath R, McMenemy N, Alkaitis M, DeLeo J: Differential migration, LPS-induced cytokine, chemokine, and NO expression in immortalized BV-2 and HAPI cell lines and primary microglial cultures. J Neurochem 2008, 107:557–569.
- Hilliard A, Mendonca P, Soliman KFA. Involvement of NFƙB and MAPK signaling pathways in the preventive effects of Ganoderma lucidum on the inflammation of BV-2 microglial cells induced by LPS. J Neuroimmunol. 2020 Aug 15;345:577269. doi: 10.1016/j.jneuroim.2020.577269. Epub 2020 May 26. PMID: 32480240; PMCID: PMC7382303.
- Hou RC, Wu CC, Huang JR, Chen YS, Jeng KC. Oxidative toxicity in BV-2 microglia cells: sesamolin neuroprotection of H2O2 injury involving activation of p38 mitogen-activated protein kinase.
- Mendonca P, Taka E, Bauer D, Reams RR, Soliman KFA. The attenuating effects of 1,2,3,4,6 penta-O-galloyl-β-d-glucose on pro-inflammatory responses of LPS/IFNγ-activated BV-2 microglial cells through NFƙB and MAPK signaling pathways. J Neuroimmunol. 2018 Nov 15;324:43-53. doi: 10.1016/j.jneuroim.2018.09.004. Epub 2018 Sep 11. PMID: 30236786; PMCID: PMC6245951.
- Cobourne-Duval MK, Taka E, Mendonca P, Soliman KFA. Thymoquinone increases the expression of neuroprotective proteins while decreasing the expression of pro-inflammatory cytokines and the gene expression NFκB pathway signaling targets in LPS/IFNγ -activated BV-2 microglia cells. J Neuroimmunol. 2018 Jul 15;320:87-97. doi: 10.1016/j.jneuroim.2018.04.018. Epub 2018 May 4. PMID: 29759145; PMCID: PMC5967628.
- The labels in the figures are confusing. Please update the figures to be clearly understood. Also, include both tails of the error bars indicating standard deviation. Failure to do so complicates the interpretation of certain figures. eg. Fig 8A: the authors claim the normalized expression of HMOX1 mRNA in Hesperitin-only and LPS-only treated cells is not significantly different. But the differences in the bars suggest otherwise. Only the upper tail of standard deviation error bar is indicated here. Similarly, the authors claim there is a significant difference between the cells treated with LPS only and Hesperitin + LPS, but there is no error bar for that treatment, and Hesperetin-only treatment and LPS+hesperetin treatments show expression at the same level. Hence, the figures need a thorough revision.
Response: As requested by the reviewer, we updated all the labels in the figures and included both tails of the error bars.
The statistical analysis was reviewed and corrected for the results of normalized expression of HMOX1 mRNA in Hesperitin-only and LPS-only treated cells. Now, the description of the results for HMOX1 is consistent with the data on the graph, and there is a statistically significant difference between Hesperitin-only and LPS-only treated cells.
The error bar for hesperitin + LPS is present in the graph; however, it is too small and sits on the top of the bar.
We agree that Hesperitin-only treatment and LPS + hesperitin treatments show expression at the same level. We suggest that hesperetin reverts or inhibits the effect of LPS, and because of it, the combination of the treatments shows the expression of HMOX1 at the same level in both treatments.
- What is the rationale behind including 24 hr treatment time point? It seems to me that most targets that are being investigated are showing the expected trend at 48 hrs. Maybe the 24-hour time point treatments can be included as a supplemental figure?
Response: Initially, 24-h studies were conducted to investigate the compound effects on the BV-2 microglial cells. However, for the antioxidant assays, there was a reduction in their activity after 24h, which was opposite to our hypothesis. Based on this, we decided to extend the treatments to 48h, and we could see an opposite, beneficial effect. We believe that 24-h hours was insufficient for hesperetin to exert its antioxidant effects. After the repetition of assays with treatment for 48-h, there was a significant increase in the antioxidant mediator activity, consistent with the literature. Therefore, we believe it is important to show the behavior of the compound at 24h and 48h.
- The authors have not explained their rationale behind choosing BV-2 murine cell line for these experiments. There are other human microglial cell lines available (e.g., HMC3 cells) that would have given far more clinically relevant results. Do the authors intend to perform in vivo mouse studies to investigate further the effects of Hesperitin in preventing neurodegeneration? Ultimately unless going towards validating these findings in an animal model, the use of this cell line may be a major limitation for this manuscript. Is any information available on the relevance or similarity between the secretome of BV-2 murine microglial cells and in vivo human microglial cells?
Response: We appreciate the question and concern. Murine BV-2 microglial cells have been used widely across research, especially for investigating neurodegenerative disorders involving immune responses, such as neuroinflammation and oxidative stress. This is an alternative model to the low cell number and time-consuming techniques required to grow primary microglia cultures. BV-2 cells proliferate and are an excellent option to yield a large number of cells quickly (1, 2).
BV-2 cells were developed by Blasi and colleges (3) through retroviral transduction in 1990 and since then, these cells have been largely used. Studies comparing primary rat microglia to the BV-2 cell line observed that upon LPS stimulation, BV-2 cells secreted less but still substantial amounts of NO compared to primary microglia (4). Henn et al. (5) investigated the BV-2 cells as an appropriate alternative to the primary cultures. They found that in response to LPS, 90% of genes induced in the BV-2 cells were also induced in primary microglia, indicating that this is a good research model.
The use of microglia cell line speed up research investigations and decrease the need for continuous cell preparations and animal experimentation, provided that the cell line reproduces the in vivo situation of primary microglia. Based on this literature, our lab has been using this model for the last decade and publishing articles in peer-reviewed journals, indicating this model is appropriate to show the potential of natural compounds as neuroprotective agents against neurodegeneration (6, 7, 8).
We agree that in vivo studies are necessary and will be included in our future studies. Please see the references below.
1- Stansley, B., Post, J. & Hensley, K. A comparative review of cell culture systems for studying microglial biology in Alzheimer's disease. J Neuroinflammation 9, 115 (2012). https://doi.org/10.1186/1742-2094-9-115
2- Hou RC, Wu CC, Huang JR, Chen YS, Jeng KC. Oxidative toxicity in BV-2 microglia cells: sesamolin neuroprotection of H2O2 injury involving activation of p38 mitogen-activated protein kinase. Ann N Y Acad Sci. 2005 May;1042:279-85. doi: 10.1196/annals.1338.050. PMID: 15965073.
3- Blasi E., Barluzzi R., Bocchini V., MAzzolla R., Bistoni F. Immortalization of murine microglial cells by a v-raf/v-myc carrying retrovirus. J Neuroimmunol. 1990;27(2-3):229-37.
4- Horvath R, McMenemy N, Alkaitis M, DeLeo J: Differential migration, LPS-induced cytokine, chemokine, and NO expression in immortalized BV-2 and HAPI cell lines and primary microglial cultures. J Neurochem 2008, 107:557–569.
5- Henn A, Lund S, Hedtjarn M, Schrattenholz A, Porzgen P, Leist M: The suitability of BV2 cells as alternative model system for primary microglia cultures or animal experiments examining brain inflammation. ALTEX 2009, 26:83–94.
6- Hilliard A, Mendonca P, Soliman KFA. Involvement of NFƙB and MAPK signaling pathways in the preventive effects of Ganoderma lucidum on the inflammation of BV-2 microglial cells induced by LPS. J Neuroimmunol. 2020 Aug 15;345:577269. doi: 10.1016/j.jneuroim.2020.577269. Epub 2020 May 26. PMID: 32480240; PMCID: PMC7382303.
7- Mendonca P, Taka E, Bauer D, Reams RR, Soliman KFA. The attenuating effects of 1,2,3,4,6 penta-O-galloyl-β-d-glucose on pro-inflammatory responses of LPS/IFNγ-activated BV-2 microglial cells through NFƙB and MAPK signaling pathways. J Neuroimmunol. 2018 Nov 15;324:43-53. doi: 10.1016/j.jneuroim.2018.09.004. Epub 2018 Sep 11. PMID: 30236786; PMCID: PMC6245951.
8- Cobourne-Duval MK, Taka E, Mendonca P, Soliman KFA. Thymoquinone increases the expression of neuroprotective proteins while decreasing the expression of pro-inflammatory cytokines and the gene expression NFκB pathway signaling targets in LPS/IFNγ -activated BV-2 microglia cells. J Neuroimmunol. 2018 Jul 15;320:87-97. doi: 10.1016/j.jneuroim.2018.04.018. Epub 2018 May 4. PMID: 29759145; PMCID: PMC5967628.
- Limitations section is missing in the manuscript. Although the discussion section covers the significance of the findings of this paper, a limitations section is necessary to list and discuss the shortcomings of this paper and its limited scope.
Response: As requested, we included the limitations of the manuscript in the last paragraph of the Discussion as follows:
In this investigation, LPS-stimulated BV-2 microglial cells were used. This is a well-characterized model used in research, especially in the study of neurodegenerative disorders involving immune responses, including oxidative stress and neuroinflammation. Many articles described that BV-2 cells are compatible substitutes for the use of primary microglia in several experimental assays, as well as in studies of a more complex nature involving cell–cell interaction [1]. Studies described that comparing BV-2 cell lines and primary rat microglia showed that LPS stimulation caused BV-2 cells to secret lesser but still significant amounts of NO compared to primary microglia [2]. Henn et al. [1] investigated the BV-2 cells as a suitable alternative to the primary cultures. They described that in response to LPS, 90% of genes induced in the BV-2 cells were also induced in primary microglia, indicating that this is a good research model. However, further studies will be needed using in vivo models to validate and elucidate the molecular mechanisms used by hesperetin to fight oxidative stress and inflammation.
- Henn, A.; Lund, S.; Hedtjarn, M.; Schrattenholz, A.; Porzgen, P.; Leist, M. The suitability of BV2 cells as an alternative model system for primary microglia cultures or for animal experiments examining brain inflammation. ALTEX 2009, 26, 83–94. https://doi.org/10.14573/altex.2009.2.83.
- Horvath, R.; McMenemy, N.; Alkaitis, M.; DeLeo, J. Differential migration, LPS-induced cytokine, chemokine, and NO expression in immortalized BV-2 and HAPI cell lines and primary microglial cultures. Neurochem. 2008, 107, 557–569.